# Utilization of 3D Scan Data: "Representation" of Korean Wooden Architectural Heritage

**Jae-Young Lee [1] and Dai Whan An [2,*]**

1   School of Architecture, Hongik University, Seoul 04066, Republic of Korea; ljy21@hongik.ac.kr
2   Department of Architecture, Chungbuk National University, ChungDaeRo 1, Seowon gu, Cheongju-si 28644, Republic of Korea
*   Correspondence: an4229@cbnu.ac.kr

**Abstract:** This study examines the characteristics of 3D scan data of Korean wooden architectural heritage structures and methods of applying the data from a representation perspective. For the preservation of its unique features, architectural heritage requires accurate representation in terms of investigation, research, and application. The representation of architectural heritage indicates a derivative reflecting the original structure and construction based on the intention of the builder. Thus, the recognition of architectural heritage varies according to the representation methods, which include text, drawings, plans, photos, and 3D scan data, each of which has different advantages and disadvantages. Compared with other models, 3D scan data provide a more accurate representation and carry value as digital data, while minimizing potential intentionality. Three-dimensional scan data can be collated with different types of information, such as building information modeling (BIM), with ease of updating. Therefore, the characteristics of 3D scan data should be identified and applied regarding representation in order to maximize data effectiveness. Three-dimensional scan data can be used most efficiently to create investigative reports representing architectural heritage. By adding the awareness of new technologies, Korean wooden architectural heritage can take advantage of accurate documentation and digital surveys in order to be preserved and recorded sustainably through digital towns and datasets. They can documented and represented through a technology-driven investigative report.

**Keywords:** representation; wooden architectural heritage; 3D scan data; accurate shape; investigative report; building information modeling; sustainability

## 1. Introduction

Representation cannot be fully described by its lexical meaning of reshowing and reproducing the original. Representation results can vary according to the intention of the producers who executed the representation and the various production methods. Diversity indicates that accuracy and value can differ [1]. Accordingly, the concept of 'representation' has been regarded as a significant issue, not only in aesthetics and literature, but also in construction [2,3].

Indeed, "representation" is important in architectural heritage. The reasons for this include, first, only one architectural heritage exists, it is the only object, which is an immovable heritage. Second, its large scale makes it difficult to recognize instantly, and therefore researchers have difficult to study architectural heritage. Additionally, managing architectural heritage, such as repairing and restoring or actively utilizing it, is strenuous; that is, there is a difficulty in approaching the "origin." Therefore, studying or using "represented" data is much easier. Efforts were made to reduce the gap between the "original," the architectural heritage, and the "represented" data. As a result of these efforts, investigative reports have been developed, and new techniques have been applied as tools for 'representation'.

Korean wooden architectural heritage is an East Asian architectural type that shows the characteristics of using wooden structures. First, as wood is used, its shape is easily changed through many repairs and restorations, and it is sometimes relocated. Second, wood is mainly used as a material, but various materials such as tiles, soil, and stones are also used. Third, the shape is complex and hard to understand. Therefore, creating represented data is difficult, and the gap between the "original" architectural heritage and the "represented" data is large [4,5].

Recognizing buildings as cultural properties and preserving their value has been practiced since ancient times in Korea. The modern concept of conservation has been studied extensively since the Republic of Korea was established in 1945. Several experts have obtained data on different architectural heritage types [6]. Therefore, various attempts have been made to research, manage, and utilize the "represented" data [5,7].

Recently, diverse techniques have been applied to ensure the highest data accuracy and multiplicity levels. Currently, photogeography, 3D scanning, etc., are used. A frequently used method is the use of a 3D scanner that obtains and applies 3D scan data of architectural heritage to identify the 3D scan data value [8]. For example, in 2008, when Sungnyemun Gate, which is Korea's National Treasure No.1, was severely damaged by arson, it was possible to restore it using the investigative report that was made in 2005 as basic data. In particular, the 3D scan data that were created during the production of the investigative report played a significant role in the restoration [9].

The results that are obtained by investigating and researching heritage based on these techniques are not only used to prepare investigative reports as basic research data but are also utilized in various directions. In other words, applying the results of research studies on architectural heritage in different directions is the most efficient method, considering that the intention of conserving the unique architectural heritage collides with that of applying it. This study also intends to make a theoretical generalization contribution to this field [5].

As architectural heritage is unique and has to be conserved as a cultural asset, changes to heritage structures cannot be allowed, rendering heritage study difficult. Nonetheless, many people are interested in applying architectural heritage styles to their property for wealth or fame. The most efficient approach is to use "represented" data based on the results of architectural heritage studies instead of using the "original" architectural heritage structures, as the actual features may prove difficult to apply. For example, we cannot conduct research on the actual painting of Mona Lisa and have no choice but to study the Mona Lisa replica, texts, and photographs. Several attempts have been made to represent architectural heritage. As another example, there have been attempts by researchers to upload and share many materials, such as 3D scan data on sites such as CyArk [10]. This has made it possible to carry out research on many cultural heritage sites that cannot be visited in practice, and 'represented' data are a good material that can be used in various ways. In this way, it is possible to conduct research on the "original" architecture using 'represented' data of higher quality.

Meanwhile, studies have examined the conceptual difference between actual and represented architectural heritage data [11]. This research can be said to be a consequential response to the efforts to create 'represented' data that are as close to the 'original' architectural heritage as possible. In other words, since the 'original' architectural heritage and the 'represented' data cannot be the same, it is important to universalize the conceptual understanding of the difference. The present study examines the characteristics of the 3D scan data of architectural heritage based on the concept of represented architectural heritage by comparing it with other data. This is because, by doing so, a more accurate understanding of the architectural heritage can be obtained, and the sustainability of the architectural heritage can be secured by confirming its value.

The Unhyeongung Royal Residence was considered to be the best case study for comparative analysis because various measured data have already been collected on it, among other Korean wooden architectural heritages. Appropriate data have been established in

order to compare and verify the architectural heritage and the Unhyeongung Royal Residence. Additionally, the experts believed that identifying similarities and differences through these comparisons would be easy. Through the case of the Unhyeongung Royal Residence, we thought it would be possible to more clearly grasp the unique characteristics of Korean wooden architectural heritage from the point of view of 'representation' [5,7].

This study proposes a conceptual method for applying the 3D scan data of represented architectural heritage. It aims to identify the characteristics of 3D scan data as represented data and presents other applicable methods. Thus, this study examines the general concept and characteristics of architectural heritage representation and identifies its importance. Moreover, this study tries to analyze the types and characteristics of representation methods, including 3D scan data, and their respective advantages and disadvantages.

The breadth of the argument is limited since all of the data types or areas cannot be handled. This discussion may also be limited to relevant experts. However, by comparing and reviewing the existing data and redefining the 3D scan data concept, the perspective of research, the management, and the utilization could be expanded.

This study has the following ultimate purposes:

First, new and existing technologies should be used simultaneously to investigate and research cultural heritage. As new attempts are crucial, re-verifying new technologies after a certain period and establishing a theoretical system through comparison with existing technologies is necessary. New technologies can then be used universally while securing reliability and stability. Second, as the 3D scan data method constitutes a new technology that has been recently applied to cultural heritage, no attempt has been made to re-verify it. This study establishes a theoretical system of 3D scan data by comparing them with similar technologies based on the philosophical concept. Third, this study aims to secure long-term stability, sustainability, and generalization through comparison with other data types in terms of 'representation' for 3D scan data, which are already widely used. Fourth, this study was intended to show that it can be used as a necessary tool for ensuring sustainability and the unique characteristics of Korean wooden architectural heritage.

## 2. 3D Scan Data and the "Representation" of Architectural Heritage

### 2.1. Concept and Purpose of "Representation"

The Korean term "Jaehyeon ( 再現)" can be translated into "representation" in English. In architectural heritage, representation first indicates a structure substituting the original; for example, a model or a drawing that can replace the original. Second, it signifies a reaction to an object presentation that is not visible at that moment. One might imagine the original cultural property by reviewing an investigative report. Third, it indicates the copy, reproduction, and development of an object or structure that previously existed, or still exists, through symbolization [1]. Therefore, the represented object is inevitably compared with and derived from the original but is considered to be not perfectly identical to the original one.

Practitioners of "representation" are classified into those who build the architectural heritage, those who produce the represented heritage, and those who accept the represented buildings [12]. Thus, these three types inevitably have different intentions and ideas on the same architectural heritage. Since architectural heritage buildings were built a long time ago, it is difficult to know the intentions and ideas of the founders, such as the designers and architects. Therefore, 'represented' architectural heritage is mostly based on the 'origin' heritage but has various perspectives that determine the value of the architectural heritage. Those who apply and consume aspects of the architectural heritage based on the represented building can only perceive the represented building rather than the "origin" building.

The typical representation methods of architectural heritage include text, image, plan, photo, and 3D scan data, as well as X-ray images, non-destructive inspection, and component analyzer data. Text and drawing data have been frequently used since ancient times and are still accountable for the largest proportion of representations. With new

techniques, including photography, representation methods have become diverse. In particular, 3D scan data that are obtained with 3D scanners have brought the most significant change in representation.

The outcomes of representing architectural heritage are presented through various methods, such as drawings, documents, and models. The investigative report is the most significant document among them as it expresses the entirety of the architectural heritage. Therefore, the original structures do not need to be directly approached or investigated. In other words, the investigative report most similarly represents the "original" architectural heritage and helps experts to research, manage, and utilize it efficiently. The investigative result, or the represented outcome, can easily make its readers aware of the architectural heritage value. Consequently, the methods, the intentions, and the outcomes of representing architectural heritage and the recognition of represented architectural heritage vary.

The recognition of represented architectural heritage varies from person to person because people choose the method to represent architectural heritage in their own unique way. Those who apply and use the represented forms perceive them differently. For example, each person has written about the Parthenon uniquely. Readers study such accounts and perceive the Parthenon similarly to the represented Parthenon. However, without seeing the building for themselves, they can only apply and use the information that they have gained from the documents on the Parthenon instead of the original structure. In other words, the understanding of 'represented' architectural heritage is inevitably different for each person. Therefore, the concept and method of 'representation' is important, and a more accurate and universal concept of 'represented' architectural heritage is needed.

### 2.2. Characteristics of Korean Wooden Architectural Heritage and the Necessity of Representation

This section deals with the characteristics of Korean wooden architectural heritage from the representation perspective and presents the importance of representation with 3D scan data.

The concept of architectural heritage can be used in various ways. Whereas in a narrow sense, it means only the building itself, in a broader sense, it includes sociocultural behaviors and perceptions that are revealed in the buildings. In this study, architectural heritage is based on data that were obtained via 3D scanning that represent the form of a building. Afterward, data expand to form an investigative report, including writings, photos, drawings, and building information modeling (BIM), from which the basic information about architectural heritage can be gathered. The investigative report is considered to be a "representation" tool. Experts in the cultural heritage field engage in research, maintenance, and utilization using the "represented" investigative reports instead of the architectural heritage itself, which is the "origin". Therefore, the characteristics of Korean traditional wooden building cultural heritage, including the characteristics of the architectural cultural heritage, are as follows:

1. Limitation of recognition: Buildings are hardly ever seen instantly with the naked eye because they are generally bigger and taller than people. In particular, people cannot easily recognize an entire building's composition, as its space is divided internally and externally. These parts should be comprehended through various representation methods, such as an expert's explanation. In other words, architectural heritage can be efficiently recognized through representation.

2. Value of conserving unique cultural properties: Architectural heritage is unique and valued as historical cultural property. Specifically, the historical and cultural value of architectural heritage should be verified through both the actual and the original architectural heritage, as well as other data, such as text and drawings of such structures. One should identify and note the historical and cultural value of architectural heritage by examining and studying it. Accordingly, the public can recognize the value of architectural heritage based on the representational data. In addition, they

can accept that the original heritage should be conserved as cultural property rather than applied in ordinary architectural practice.

3.　Difficulty of approach and active utilization: Approaching architectural heritage tends to be controlled because heritage should be conserved through maintenance. The active utilization of original architectural heritage is limited, although it is emphasized more than it was in the past because its conservation is more important than its utilization. In this sense, using represented data can increase the possibility of approaching heritage and more easily applying architectural heritage features.

4.　Characteristics of Korean wooden architecture: First, as mentioned earlier, wood was primarily used in Korean architectural heritage, the shape of which can be easily altered through many repairs and restorations, and it can sometimes be relocated. Accordingly, charters such as "The Nara Document on Authenticity (1994) [13]," "Principles for the Preservation of Historic Timber Structures (1999) [14]," and "Principles for the conservation of wooden built heritage (2017) [15]" have been created. Second, apart from wood, other widely used materials were roof tiles, soil, and stones. Third, the shape is complicated to understand. Therefore, creating "represented" data is difficult, and the gap between the "original" architectural heritage and the "represented" data is large.

5.　Timber transforms more significantly than other materials in heritage structures and its shape and location can change more easily. Especially in Korean wooden architecture, small components of varying shapes are combined. Therefore, many parts are difficult to represent using existing representation methods.

Maintaining architectural heritage for conservation collides with noting and applying architectural heritage value. In addition, architectural heritage is difficult to perceive easily without represented data. Therefore, represented data should be derived in order to preserve, note, and apply the value and the characteristics of architectural heritage. In Korea, many architectural heritage structures are constructed from wood. Thus, the representation needs to be tailored to the special characteristics of wooden architecture.

As mentioned above, Korean wooden architectural heritage is difficult to represent, given the large gap between the "original" and the "represented" data pertaining to architectural heritage, making it a suitable research case study. To address this gap, an investigation of a case study was conducted in 2016, and the Unhyeongung Royal Residence, where the investigative report was released, was selected [16]. This study does not address new techniques in 3D scanning, but rather the concept of 3D scanning data. Therefore, regardless of the measurement timing, the actual measurement of the Unhyeongung Royal Residence established data that were appropriate for comparing and verifying the architectural heritage. In other words, the data of the Unhyeongung Royal Residence were properly constructed in order to directly compare 3D scan data with drawings and photos because no existing studies directly compared the "original" architectural heritage and the "represented" data.

### 2.3. Overview of 3D Scan Data

This section describes the general characteristics and the utilization of 3D scan data and the advantages in architectural heritage research.

A 3D scanner determines the point density within a certain range. The object is irradiated with a laser in order to measure the reflection of the object's surface and to obtain the coordinates of the position of a point on the surface. The collected data are called 3D scan data, or point cloud, where the points are clouded together in order to establish 3D spatial coordinates in the target object shape. This point has no size, only position coordinates. The 3D scan data size is determined by the number of points. Although 3D scan data files tend to be large in size, they can vary depending on the point density of the scan target.

In general, a dedicated computer program is used to view the scan data obtained with a 3D scanner. Based on this program, a single dataset is created by pasting multiple cuts of data obtained in the field. Noise cancellation is performed to remove unnecessary

parts. This procedure transforms a dataset into a complete point cloud. Then, as the 2D photo data are piled up with the 3D scan data, the 3D scan data representing the color that is most similar to the original building are created. Accordingly, due to the density of the point cloud, the 3D scan data method is more accurate and unintentional than the other representation methods. In addition, 3D scanning uses 3D and digital data. Data, including three-dimensional forms and locational information, can be stored and analyzed through this process, as shown in Figures 1–8. Figures 1–8 show various types of data on Unhyeongung. Among them, 3D scan data have the advantage of being able to show the shape and location information most accurately. This advantage can be confirmed because a lot of various types of data on Unhyeongung were already built in 2016. However, 3D scan data technology is not widely available because of the high cost of the scanning equipment, the complex operating programs, and the large amount of data.

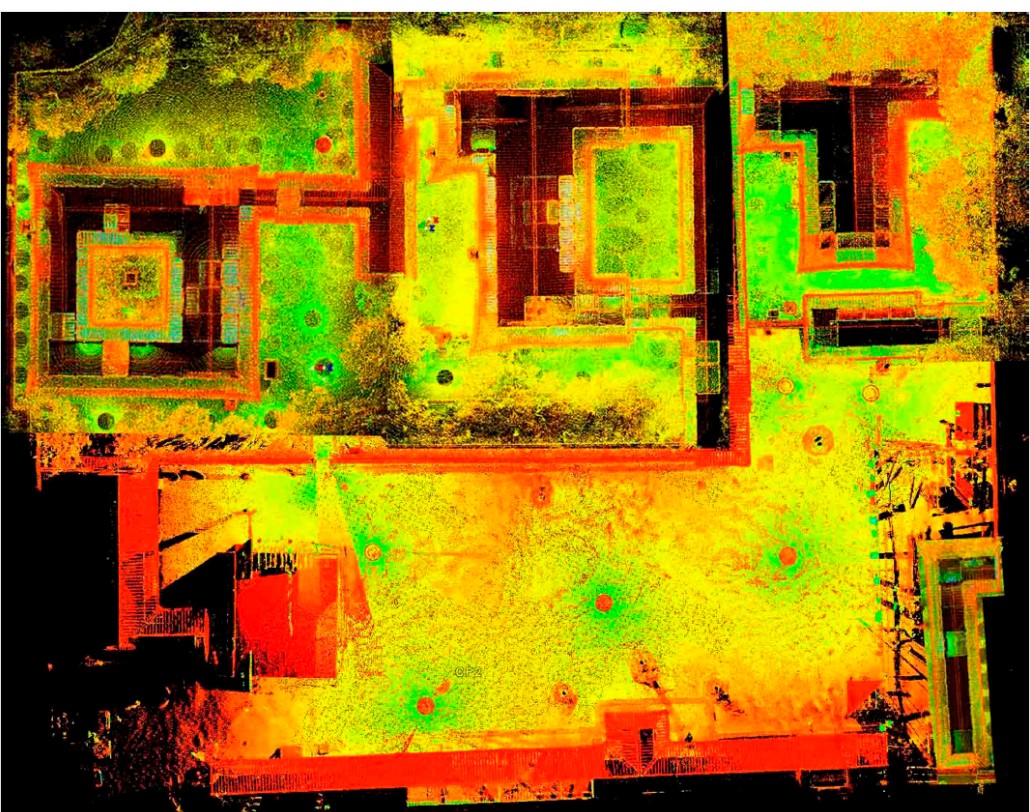

**Figure 1.** A 3D scan data of Norakdang in Unhyeongung.

As technology advances in the future, the 3D scan data method could be further developed along with modeling and used effectively. A 3D scanner extracts data using professional equipment in the professional industry; thus, its generalization requires the availability of ordinary equipment to the general public. With advanced technology, this availability is expected to become more common, and the concept of 3D scan data may also expand.

## 1. 평면

노락당(老樂堂)은 정면 9.5칸, 측면 3칸의 —자형 몸채에 동쪽 2칸에서 남쪽으로 2칸, 서쪽 1 칸에서 남쪽으로 2칸, 동쪽 3칸에서 북쪽으로 1.5칸 내민 평면을 이루고 있다. 노락당 정면 전체 길이는 24,305mm, 측면 전체길이는 16,986mm이며, 전체 면적은 초석과 벽체로 구성된 반침을 포함하여 261.17㎡(79.00평)이다.

노락당은 남행각과 함께 ㄴ자 마당을 구성하고 있으며 북측으로는 북행각과 뒷마당을 구성한다. 동익채와 서익채는 노락당 남행각, 대문채와 각각 연결되어 있으며 서로간의 통행은 불가능하다. 노락당 북익채는 노락당 북행각과 연결되어 퇴칸으로 통행할 수 있다. 몸채는 전후면에 퇴칸을 두고 중앙 3칸을 넓은 대청으로 사용하고 있다. 대청과 서측방 1칸의 전면 퇴칸은 통로로 사용되고 있으며 나머지 서측방 1칸과 동측방 2칸의 전면 퇴칸은 창호를 달아 마루방으로 꾸몄다. 대청과 동측방 1칸의 후면 퇴칸에도 마루를 두어 통행하였다. 대청 양 옆으로는 방을 2겹으로 구성하고 동·서단에 부엌과 방을 구성하였다. 부엌 상부에는 다락방을 두었다. 동·서익채의 끝칸은 쪽마루를 둔 온돌방이며, 뒷마당과 면한 북측면에도 쪽마루를 설치하였다. 배면 회첨부의 쪽마루는 지붕아래까지 벽체를 구성하여 내부 공간을 확장하였다. 벽체가 없는 배면과 동측면 쪽마루에는 난간이 설치되었다. 이러한 전후퇴의 구성과 중앙 3칸의 대청 동서양측으로 온돌방을 2칸씩 배치하는 구성은 궁궐내전의 평면구성방식과 그 맥을 같이하는 것이다.[1]

**Figure 2.** Representation with text: Explanation of the floor plan of Norakdang in Unhyeongung in an investigative report (in Korean characters as text).

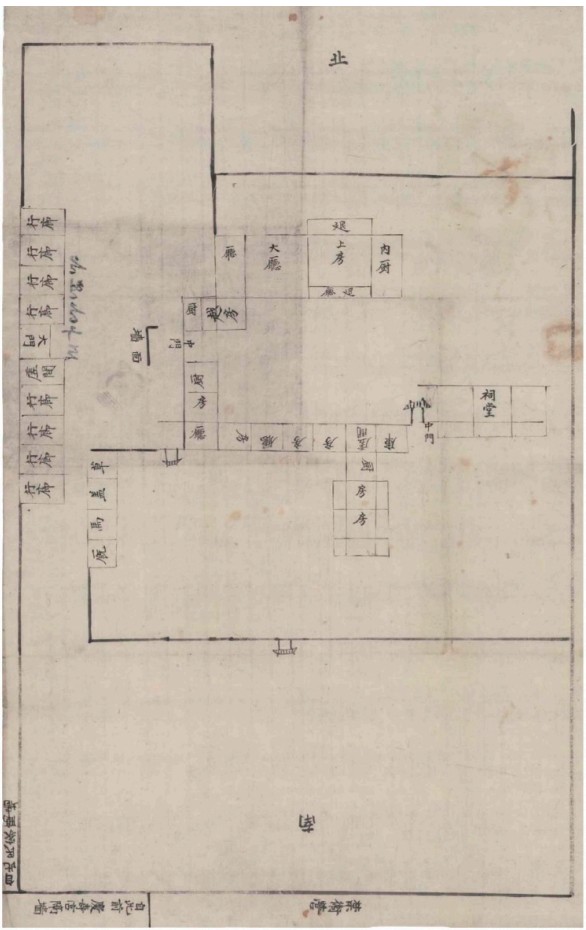

**Figure 3.** Representation with drawings (1863): Norakdang in Unhyeongung.

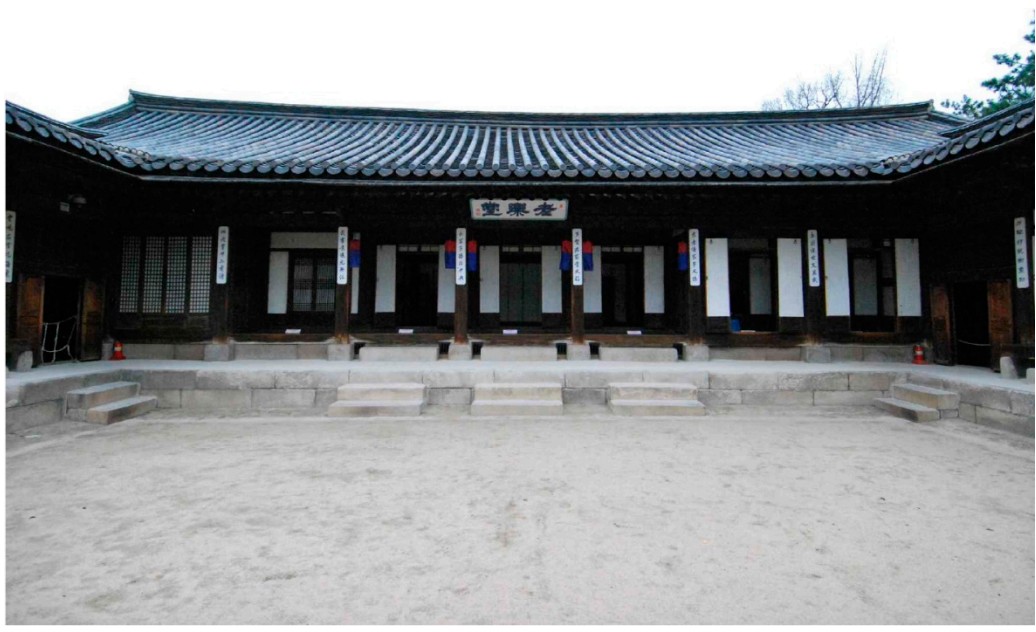

**Figure 4.** Representation with photo: Norakdang in Unhyeongung.

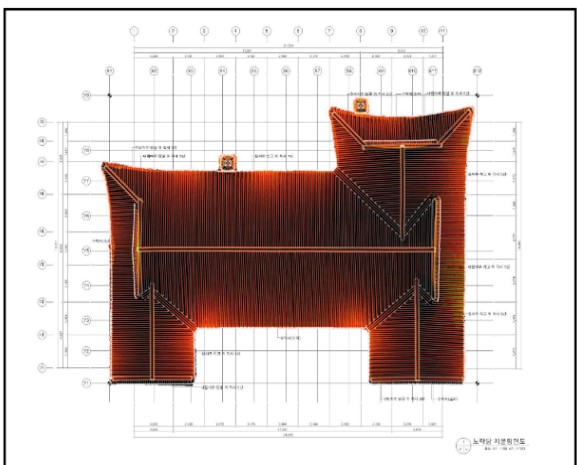

**Figure 5.** Roof of Norakdang in Unhyeongung: 3D scan data 100%, drawing 0%.

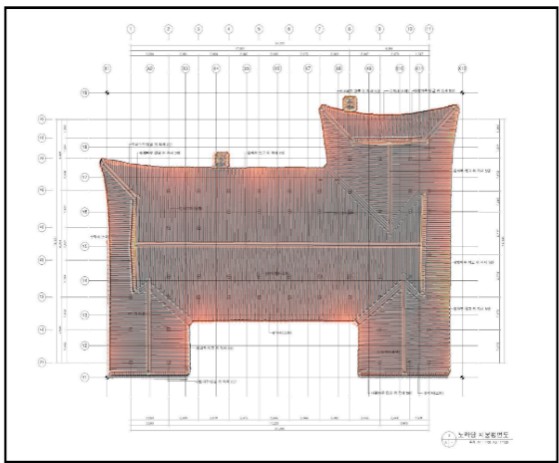

**Figure 6.** Roof of Norakdang in Unhyeongung: 3D scan data 50%, drawing 50%.

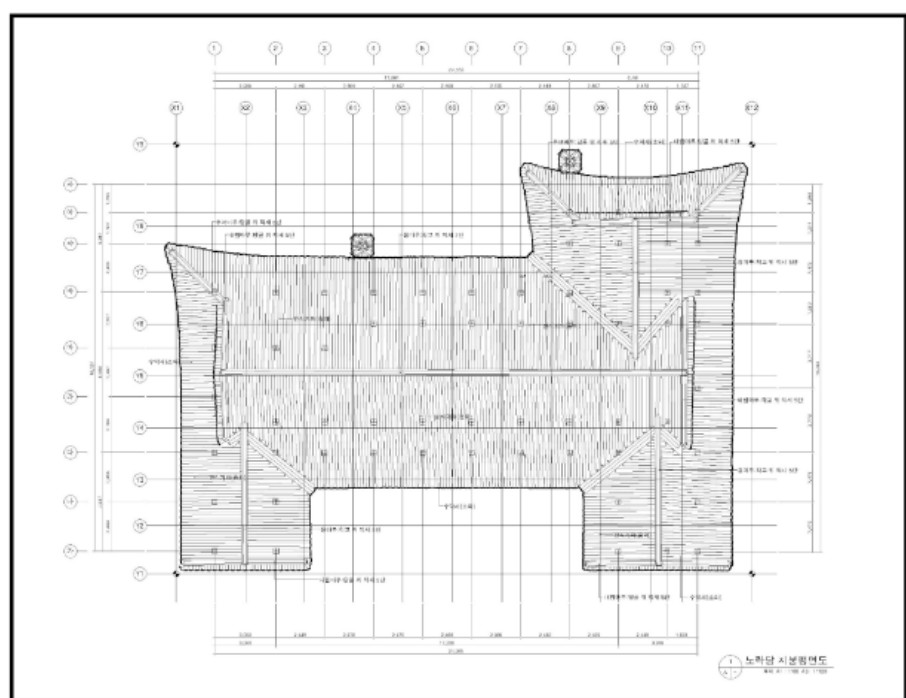

**Figure 7.** Roof of Norakdang in Unhyeongung: 3D scan data 0%, drawing 100%.

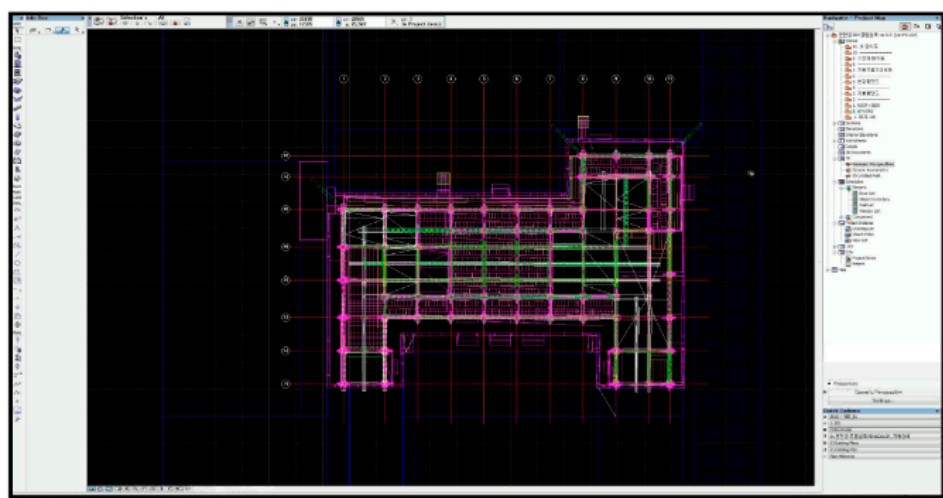

**Figure 8.** Upper wooden structure BIM of Norakdang in Unhyeongung.

## 3. Characteristics of 3D Scan Data from the Perspective of Representation

This section examines the characteristics of the representation methods of architectural heritage, such as text, drawing, plan, photo, model, and 3D scan data, from the representation perspective. As 3D scan data use has been recently introduced and has not been evaluated properly, this section provides its comparison with other methods in order to reveal the pros and cons of each and thereby offer insight into its effective use. The methods are compared regarding accuracy, intentionality, and utilization. Three-dimensional scan data show the pros that compensate for the drawbacks of the other representation methods.

### 3.1. Accuracy

Drawings and text-based reports have been conventionally used to represent architectural heritage for as long as history has been recorded. For example, a hieroglyph containing "palace" and "temple" can shape the perception of the represented architecture according to the person perceiving the representation [17]. Text was the primary representation method before modern times. As it does not visually show the form of structures, a certain degree of imagination is needed. Moreover, as the text selectively represents parts of the actual architectural heritage, it has a significantly low accuracy as a method of representing architectural heritage. However, it remains an important representation method due to its advantages. First, the history, correlation, connotation, and symbolism of architectural heritage can be represented only through text. Second, a textual representation allows for an extensive explanation, comparison, and analysis. Third, the size and the volume of structural spaces can be accurately presented through the expression of numbers.

Drawing, such as a wall painting in a cave, is the oldest representation method. With drawings, the characteristics of architectural heritage can be instantly recognized, compared to textual representation. Thus, drawing constitutes a more accurate representation method than text regarding the morphological aspects. However, drawing can also be considered to be a partial "representation" in that the artist selectively draws parts of practical heritage structures.

Drawing as a representation method is classified from the modern perspective into general drawing that the public can easily understand and the drawing plan that requires architectural knowledge for interpretation. The plan is a special representation method that is more accurately detailed compared to the general drawing method and helps specialists to recognize the structure and shapes of the architectural heritage precisely. Therefore, combining text and plan representations, which complement each other, has been used as an effective method.

Photos representing architectural heritage have ensured accurate representation regarding form and color. A photo can almost perfectly represent the parts that otherwise cannot be expressed through text or drawing plans within the 2D frame. However, the accurate sizes of structures are hardly identified, and the form and color are likely to be distorted. Nonetheless, photos are increasingly used as special photography techniques, and color calibration functions provide a higher similarity between the actual architectural heritage structure and the represented one.

Meanwhile, 3D scan data are the latest tool of representing architectural heritage. As examined in Section 2.3, this method varies from existing methods in that a dense point cloud establishes the architectural heritage form on a computer. Every point contains 3D information on the heritage structure. Therefore, the 3D scanning method accurately visualizes the architectural heritage form. In addition, similar to digital photos, data representing the most accurate form can be easily copied. A particular limitation of 3D scan data and photos is the inability to express the parts that are represented only through text, as shown in Figure 1.

### 3.2. Intentionality

One of the most important factors in representing objects is intentionality [18], which indicates the methods that have been used by a person producing the representation to represent selected parts of the architectural heritage. It also determines whether the represented outcome reflects the producer's intention. The represented outcome in the text and drawing is derived as intended by the person conducting the representation when the entire architectural heritage structure cannot be accurately represented. In other words, representation inevitably includes the intention or ideas of its producers.

The photo and 3D scan data methods represent what the producer intended during the heritage representation. For these methods, intentionality cannot be included in the data that are obtained with range selection. However, both methods differ significantly: the photo has a better outcome when post-processing, through which intentionality is in-

dicated and not performed, whereas 3D scan data can represent the entire form of the architectural heritage more accurately when post-processing is carried out. In other words, considerable data can be obtained, and intentionality can be removed through scanning at several stages. Thus, the text, drawing, photo, and 3D scan data differ in their intentionality. As 3D scan data include little intentionality, it is the most accurate; the representation producer cannot freely adjust the data.

### 3.3. 3D Data

Three-dimensional and two-dimensional data differ significantly. Three-dimensional data refer to the 3D space information closely matching human perception. A building can be considered as 3D information, indicating that 3D data comprise more information than 2D data. Only 3D scan data can offer more information, and a closer match to the human perception system [19,20].

Text-based representation does not provide a visual dimension, as it requires imagination based on an accurate explanation. Drawing is generally 2D but can be represented as 3D in the case of perspective drawing. Photos represent 2D data, but with specific equipment, such as 3D photogrammetry, they can appear in 3D.

Although drawing and photo representations merely appear in 3D, specific 3D information cannot be obtained through these methods. Three-dimensional information can only be identified on a 2D monitor through 3D scan data, which facilitates 3D information verification on the entire architectural heritage form and individual points. Three-dimensional scan data are distinguished from text, drawing, and photo representations in that the structure is represented as 3D information.

### 3.4. Digital Data

As text and drawings in the traditional method are not digital data, the represented data cannot be identically copied, adjusted, or applied. Currently, most text, pictures, and photos are expressed as digital data. Nevertheless, in text, picture, and photo "representation", the "origin" is essentially the same. Accordingly, even if digital data can be copied, adjusted, and utilized, the different nature of these data does not change from the "representation" perspective.

However, 3D scan data are essentially digital in nature. Digital data have the following characteristics [5]: First, the represented data can be copied rather than reproduced or recreated and duplicated infinitely. Thus, digital data can produce and spread represented data with the same accuracy. Second, digital data representations have high value, particularly high accuracy, when they are used on a computer. As a photo provides 2D data, the data shown on the computer slightly differ from the output. Meanwhile, 3D scan data, as 3D data, can be accurately confirmed only on a computer. Third, unlike the text and drawing methods, the digital data application can be easily adjusted and complemented. This applicability increases as the represented data are processed. Last, as information can be added to 3D scan data, it can provide the most accurate representation and can accommodate additional information, such as digitized text descriptions and drawings. Table 1 briefly describes the characteristics of each method.

**Table 1.** Characteristics of each tool for representation.

|  | Accuracy | Accidentality | Dimension | Digital Data |
|---|---|---|---|---|
| Text | Very low | Intentional | No-dimension | Analog |
| Drawing | Lower | Intentional | Two-dimension | Analog |
| Digital photo | High | Accidentality | Two-dimension | Digital |
| 3D scan data | Very high | Accidentality | Three-dimension | Digital |

## 4. Application Value of 3D Scan Data of Represented Architectural Heritage

### 4.1. Investigative Report

An investigative report is generally prepared to represent the architectural heritage of the structure precisely. The investigative report uses all of the representation tools, including text, drawings, photos, 3D scan data, and other data. Moreover, the outcomes that are most appropriate for the representation are derived.

Although 3D scan data have been applied in investigative reports on architectural heritage for nearly 10 years [7], they have not been actively used for various reasons. First, 3D scanning equipment and programs have yet to be commercialized. Second, the technical links between the offices that observe and design cultural properties, those that prepare the investigative report, and firms that carry out 3D scanning are insufficient. Third, the established form of investigative reports is limited to the print format [21].

When 3D scan data are not appropriately used for the investigative report, the latter's purpose of providing an accurate representation form remains unfulfilled. Representation tools should be used according to their roles in the investigative report. Text, plan, and photo representations serve their own unique roles. In particular, both the plan and the photo methods are mainly used in the visual 2D field. An investigative report is designed to constitute both of these methods, considering their advantages and disadvantages, thus helping people to understand the represented structure. However, the role and the relationship of 3D scan data concerning the plan and photo methods are unclear. Accordingly, as plans are generally not established by referring to 3D scan data, which provides basic data, a wide gap exists between 3D scan data and the plan method.

To increase the consistency between 3D scan data and the plan method, the two representation tools can be overlapped, as shown in Figures 9–12. Through this overlap, the investigative report can represent the architectural heritage of structures more accurately. Furthermore, the form of the investigative report should be adjusted in order to reflect visual 3D data. An (2015) emphasized this issue in his study on the value of 3D data, which cannot be identified in the form of books [21].

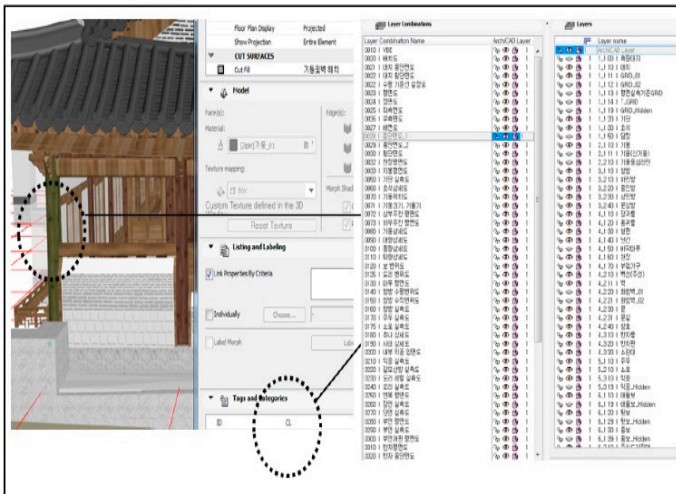

**Figure 9.** ID and layer in the BIM of Norakdang in Unhyeongung.

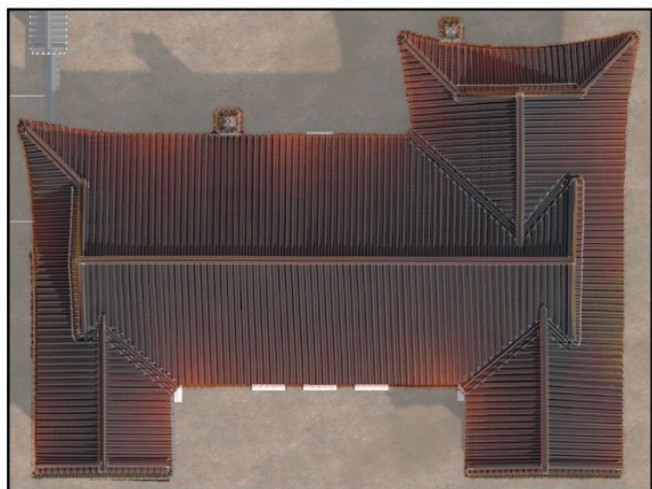

**Figure 10.** Roof of Norakdang in Unhyeongung: 3D scan data 50%, BIM 50%.

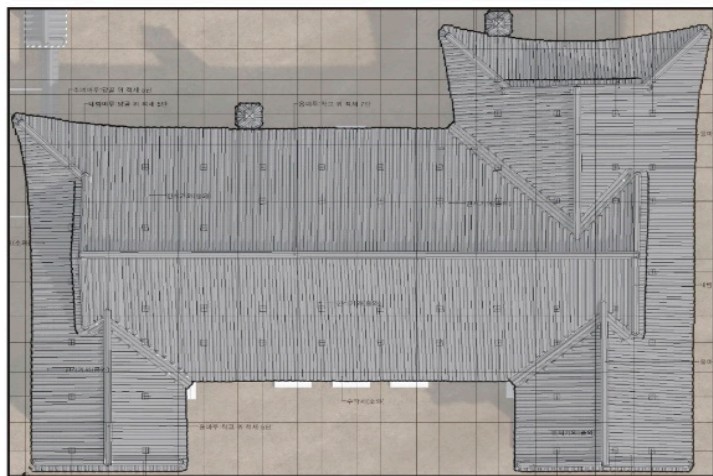

**Figure 11.** Roof of Norakdang in Unhyeongung: Drawing 50%, BIM 50%.

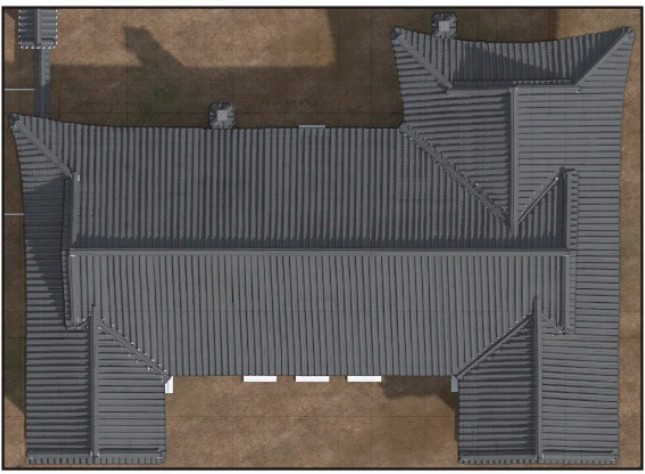

**Figure 12.** Roof of Norakdang in Unhyeongung: Drawing 0%, BIM 100%.

*4.2. Building Information Modeling (BIM)*

Only 3D scan data have a 3D form among the different representation tools. As they most accurately represent the architectural heritage form, applying 3D scan data to BIM can be considered. BIM is the technique of converting 2D architecture and design into 3D

and integrally utilizing the totality of the building information to produce and manage various pieces of information from the design to the maintenance. However, BIM programs representing architectural heritage are generally simple and thus completely different from the original form. Attempts have been made to increase the efficiency of maintaining and managing architectural heritage by simplifying the form. Previously, simplification techniques were considered to be essential in order to conserve and apply architectural heritage characteristics. Given this simplification, a perception was formed when the technical development had not maintained the capacity and the processing speed of computers and the development of proper programs [22]. Presently, the technical issues in processing digital data are gradually being solved; moreover, people can work with heavy files easily.

BIM has not been implemented in architectural heritage conservation, and 3D scan data have not been connected to BIM. As shown in Figures 12 and 13, combining 3D scan data and BIM can facilitate the review and the maintenance of each architectural heritage component in the morphologically accurate form in the BIM program. This advantage is attributed to the feature of 3D scan data accommodating various types of information that can also be implemented in BIM, such as the size and replacement history of components and their correlation with the other components in the 3D-represented form. Furthermore, different types of information, including the historical context and the cultural value, can be easily added to the 3D scan data. Such advantages of combining 3D scan data and BIM also ensure morphological accuracy.

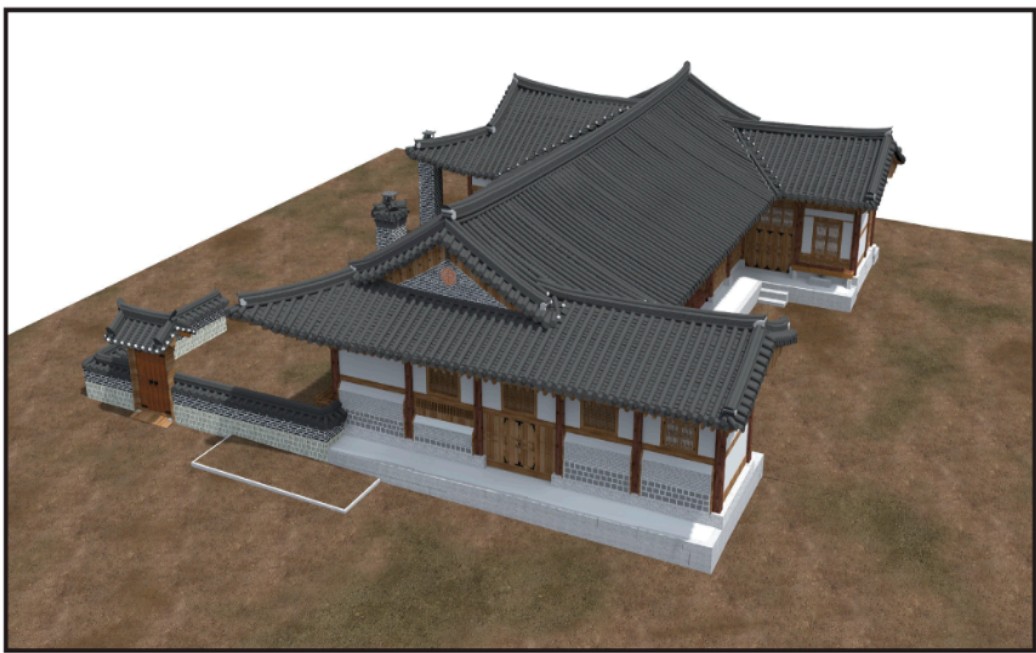

**Figure 13.** Display and BIM of Norakdang in Unhyeongung.

Recently, several attempts have been made to combine 3D scan data with BIM. Advancements in technology and equipment have enabled large-capacity processing and appropriate programs to be created, which were previously impossible. Consequently, the advantages of the morphological accuracy of 3D scan data and the usability of various pieces of information possessed by BIM are expected to create synergy.

Thus, a different type of investigative report in 3D form, which shows various aspects of the architectural heritage of structures, can be formed. In addition, the applicability of this form of an investigative report in maintaining, managing, and repairing architectural heritage is improved [22]. The trend is to go all digital and adopt 3D and integrative data on the investigative, research, maintenance, repair, and other aspects of heritage study. BIM included in 3D scan data can integrate other representation methods, such as digital data, and then apply such integrative data. In other words, BIM offers the best utility

against information loss and can control data from all of the representation methods that are relevant to heritage preservation.

### 4.3. *Application of 3D Scan Data for Producing Models and 3D Images*

#### 4.3.1. Producing Models

A definitive model for applying cultural heritage characteristics cannot be easily produced for several reasons [16]. First, the materials and the form of the model's components need to be omitted, considering that architectural heritage tends to be large, while the reduction ratio of models should be high. Second, architectural heritage components combining various materials are unlikely to be represented. Indeed, material classification is also difficult, even if all of the components are represented. Third, the precise form of a reduced model would be challenging to implement. However, combining 3D scan data and 3D printing produces a precisely represented model. This technique increases the possibility of representing and producing a detailed model for research and investigation that could not have been formed before. Therefore, in the future, efforts are needed to produce more types of data as tools for 'representation'. That way, we will be able to use a more accurately 'represented' model.

#### 4.3.2. More Accurate Video

In the past, the most accurate method of representing architectural heritage was the use of images. Photos with the same value can be arranged as a video, which can be applied as useful data to show the overall form of heritage structures. A problem with this method is that the size and the form of structures can be identified only by concerning proportions. Furthermore, the method of producing the architectural heritage form and taking a video, as shown in the existing BIM method, cannot lead to accurate representation. In this regard, 3D scan data representation remains superior. Three-dimensional scan data produce images that are distinct from those that are obtained through different methods by indicating the accurate size, proportion, and form of the structure. In addition, using bilateral information delivery media can enable users to obtain information on the accurate size and form of the architectural heritage.

## 5. Conclusions

Original architectural heritage study has some limitations because heritage characteristics cannot be easily recognized for prioritizing conservation, consequently limiting their access. Therefore, architectural heritage is represented through numerous tools for application and research. Representation indicates the creation of a product that is different from the original one, according to the producers' methods and intentions. Various attempts that have been significant in architectural heritage conservation have been made in order to ensure accuracy in representing the original form.

Recently, 3D scanning techniques have been applied as new representation tools, along with the text, drawing (plan), and photo methods as traditional representation tools, in architectural heritage. Three-dimensional scan data contain accurate information without intentionality and show unique properties as 3D digital data. As a new representation tool, their applicability value is high, considering their characteristics. First, they can show the accurate architectural heritage form in the investigative report. Second, the convergence of 3D scan data and BIM facilitates the implementation of accurate 3D information in BIM. Last, 3D scan data can be applied to produce precise models and 3D images.

This study examined 3D scan data use regarding architectural heritage representation and analyzed the 3D scan data characteristics compared to other available representation methods. Furthermore, applying 3D scan data in heritage representation has been proposed, which may broaden the applicable range of 3D scan data in architectural heritage conservation. In particular, it can be actively used in investigative reports. Three-dimensional scan data will be utilized more due to their merits and potential for application and extension in architectural heritage. Such utilization should be located and defined

as 3D scanning has to be used with other representation methods. Three-dimensional scan data will be used as the standard representation method because they extend the impacts of BIM, videos, investigative reports, and even text-based representations.

As a purpose and result of the study, in conjunction with technologies such as BIM, 3D scan data will likely be used as "represented" data that are closer to the "original." This study attempted to expand the conceptual understanding of 3D scan data regarding reproduction. This review may be controversial, however, it provides a unique perspective on the various technologies that are presently applied to Korean wooden architectural heritage.

This study analyzes the case of 3D scan data, which are already widely used but have instability, and aims to use them more universally and stably by theoretically and conceptually confirming them. The ideal development direction would be creating synergy by simultaneously using new and existing technologies. For this purpose, it is necessary to re-verify the new technologies that are already in use theoretically. Furthermore, discussing how to establish a theoretical system through comparison with existing technologies is essential. This will create an opportunity to secure sustainability and universality when another new technology is used for cultural heritage in the future.

**Author Contributions:** Conceptualization, J.-Y.L.; methodology, J.-Y.L.; software, D.W.A.; validation, D.W.A.; formal analysis, J.-Y.L.; investigation, J.-Y.L.; resources, D.W.A. and J.-Y.L.; data curation, D.W.A. and J.-Y.L.; writing—original draft preparation, D.W.A. and J.-Y.L.; writing—review and editing, D.W.A. and J.-Y.L.; visualization, D.W.A. and J.-Y.L.; supervision, J.-Y.L.; project administration, D.W.A. and J.-Y.L.; funding acquisition, J.-Y.L. All authors have read and agreed to the published version of the manuscript.

**Funding:** This work was supported by the National Research Foundation of Korea (NRF) grant funded by the Korean government (MSIT) (No. 2021R1F1A1061803).

**Institutional Review Board Statement:** Not applicable.

**Informed Consent Statement:** Not applicable.

**Data Availability Statement:** Not applicable.

**Conflicts of Interest:** The authors declare no conflict of interest.

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
