# Peer review of "Utilization of 3D Scan Data: “Representation” of Korean Wooden Architectural Heritage"

_sustainability, doi:10.3390/su15086932_

Round 1

Reviewer 1 Report

The topic is interesting, provocative and of interest to a global audience. Specifically significant is the fact that the proposed representation methodology can be applied to all types of architectural heritage, regardless of the type of construction, program typology, etc.

However, despite the clear views and theses, the paper needs further revision in order to be at the required scientific level for publication in this journal:

1)      The paper lacks a clear presentation of research methods and techniques. It is necessary to introduce  a Materials and Methods section  with sufficient detail on  the main methods applied

2)      Referencing must be improved. The paper presents a very limited number of relevant sources. Although the discussed issue is relatively new, it is necessary to refer in more detail and comprehensively to papers and studies that dealt with topics and approaches of importance to this paper, especially within the theoretical framework.

3)      It is necessary to improve the Conclusion. It currently shows the facts and views already presented in the previous text.

4)      Before line 150, it is necessary to introduce the presented points.

5)      The text is very difficult to read and follow because extensive English language and style editing is required.

Author Response

Thank you for reviewing this paper.

Detailed responds are included in the attached file.

Thanks for pointing me in the right direction.

Reviewer 2 Report

Dear Authors,

The manuscript is original and has potential. However, in my opinion it must be improved to your information can reach a large audience of readers.

Literature review/ state of the art must be revised, section 2.1:

-          - Justify with references between lines 36 to 42.

-      -    What is “original architectural heritage” and “represented data”, clarify these two concepts?

-     -     Justify with references between lines 44 to 47.

-         - Which techniques do you speak between lines 48 to 49.

-       -    It is too soon to this conclusion (lines 51 to 55). Introduce references to justify.

-        -  Many general considerations without a theoretical background/ references to support the future outcomes, when exposing the pertinence of this thematic.

-         -  In general, the English language can be improved to better make the readers understand the ideas of the authors’ exposition.

-         - Explain the difference and advantages of the new technologies when confronted with the old technologies (give some examples/ references), lines 73-82.

-         -  Explain which is the “universality” in this context, lines 90 to 91.

-        -  Explain better this sentence, lines 105 to 106.

-          - Explain “origin heritage”, line 107.

-          - To argue what is written, in lines 128-134, you should explain the previous concepts that it was recommended by the reviewer. Only with clear and contextualized concepts the author will have success in their exposition.

Section 2.2:

-       -   Introduce a reference to justify lines 140 and 141.

-         -  Introduce the aims of the five items explained in this section (lines 150 to 183).

-          - Insert references to the charter and recommendation, lines 171 to 179.

Section 2.3:

-        -   More detailed information should be given to the figures, and their pertinence for the article (line 221).

Section 3.1:

-        -  In section 3.1. the authors should underline the use of 3D scan or add-value of this new technique when comparing with the traditional representation of the architectural heritage in their rigorous draws. In addition, examples of the woods draw should be given, to better understand your argumentation.

Section 4.3.1:

-        -  In section 4.3.1., lines 414 to 423, the authors reach to the conclusion that these methods are not enough to have a complete model of the architectural heritage. In my opinion, they should give more solutions or future research should be displayed to reach to other solutions/ resolutions, as the authors are arguing.

I hope with these changes you could improve the manuscript.

Author Response

(The authors gave the same response as above.)

Reviewer 3 Report

A revision is in order with a fresh pair of eyes, as some parts seem to be draft elements that were not removed (e.g. the words between parentheses, lines 36-40).

Lines 58-62: this is an interesting sociological feature. Is there any literature on this? Perhaps a reference on similar realities in private and public buildings elsewhere, such as classical columns on contemporary houses?

A formal detail is the use of quotation marks to highlight or emphasize certain words, such as  “represent”, “original”, “clouded together”, among several others. I wonder if they are really needed to maintain fluidity in the text.

In terms of content, there is an outline of the features of wooden architecture, and then an overview of the potential applications of 3D scanning / BIM articulations. It must be stated that multiple papers deal with this latter part already, with very great technical detail, as can be verified even in other MDPI journals, more specialized in this field. The key originality here is the case study, the Korean specificity that is. From this perspective, I hesitate between recognizing a very nice text to read, on the one hand, and admitting that this sort of 3D data application is not as innovative as it is presented, on the other. Ten years ago, this would have been a fine paper on the generic potential of 3D, but in 2023 there are hundreds of very comprehensive studies on the subject. The text is structured as if until now only 2D models, photographs, and so forth are being used in architectural research worldwide, but this is obviously not the case. I believe in the paper, and both the general overview and some critical perspectives are OK, yet something is lacking, namely a more explicit insistence on the software, equipment, physical limitations, protocols, and/or operational methods behind what is generically called 3D scanning.

Author Response

Thank you for reviewing this paper.

Detailed answers are included in the attached file.

Thanks for pointing me in the right direction.

Reviewer 4 Report

The study presents an investigation of the 3D scan data of Korean Wooden Architectural Heritage. Especially the documentation of historical artifacts is an important research topic. However, the structure of the study is more like a report rather than a research article.

1) No reference has been made to similar studies in the literature. There are many studies on historical artifact modelling. A summary of the literature should be presented.

2) The 3D scanning technique used should be explained together with its mathematical background.

3) Accuracy analysis should be explained in detail. How it was done. It should be expressed quantitatively instead of qualitative expressions such as very low, lower presented in Table 1.

4) The original value of the work is low. Additional evaluations can be made on model details, especially on modeling details.

Author Response

(The authors gave the same response as above.)

Round 2

Reviewer 1 Report

Dear authors, thank you for the detailed answers. I recommend publishing the paper in its present form. Kind regards

Author Response

Dear reviewer,

Thank you for your review.

And we have reinforced some points of your previous review.

Best Regards,

Reviewer 2 Report

The references introduced enriched the manuscript.

The reviewer' suggestion to add "wooden draws", it was to clarify details of the “Norakdang in Unhyeongung" structure, adding value to the 3D scanning technique.

Author Response

Dear Reviewer,

Thank you for your review.

We added the reference 'the investigative report of Unhyeongung Royal Residence ' for understanding of details of the “Norakdang in Unhyeongung" structure.

Thank you.

Reviewer 3 Report

I have read the revised version and am satisfied with the changes. I continue to believe some fundamental aspects could have been written differently, but I also admit this may be said about most papers, including my own ones. In summary, it is not a groundbreaking contribution but still a good paper that deserves to be read and cited by others.

Author Response

(The authors gave the same response as above.)

Reviewer 4 Report

Revisions have been made. The study is acceptable as it is.

Author Response

(The authors gave the same response as above.)
